

# Identification of WRKY gene family in *Dioscorea opposita* Thunb. reveals that *DoWRKY71* enhanced the tolerance to cold and ABA stress

Linan Xing, Yanfang Zhang, Mingran Ge, Lingmin Zhao and Xiuwen Huo

College of Horticulture and Plant Protection, Inner Mongolia Agricultural University, Huhehaote, Inner Mongolia, China

## ABSTRACT

WRKY transcription factors constitute one of the largest plant-specific gene families, regulating various aspects of plant growth, development, physiological processes, and responses to abiotic stresses. This study aimed to comprehensively analyze the WRKY gene family of yam (*Dioscorea opposita* Thunb.), to understand their expression patterns during the growth and development process and their response to different treatments of yam and analyze the function of *DoWRKY71* in detail. A total of 25 *DoWRKY* genes were identified from the transcriptome of yam, which were divided into six clades (I, IIa, IIc, IId, IIe, III) based on phylogenetic analysis. The analysis of conserved motifs revealed 10 motifs, varying in length from 16 to 50 amino acids. Based on real-time quantitative PCR (qRT-PCR) analysis, *DoWRKY* genes were expressed at different stages of growth and development and responded differentially to various abiotic stresses. The expression level of *DoWRKY71* genes was up-regulated in the early stage and then down-regulated in tuber enlargement. This gene showed responsiveness to cold and abiotic stresses, such as abscisic acid (ABA) and methyl jasmonate (MeJA). Therefore, further study was conducted on this gene. Subcellular localization analysis revealed that the DoWRKY71 protein was localized in the nucleus. Moreover, the overexpression of *DoWRKY71* enhanced the cold tolerance of transgenic tobacco and promoted ABA mediated stomatal closure. This study presents the first systematic analysis of the WRKY gene family in yam, offering new insights for studying WRKY transcription factors in yam.

The functional study of *DoWRKY71* lays theoretical foundation for further exploring the regulatory function of the *DoWRKY71* gene in the growth and development related signaling pathway of yam.

## INTRODUCTION

WRKY transcription factors (TFs) are the seventh largest family of regulatory genes screened from higher plants (*Tiika et al., 2020*). The name of the WRKY family is derived from their conserved WRKYGQK amino acidcore sequences in the N-terminal region.

Corresponding author
Xiuwen Huo,
huoxiuwen@imau.edu.cn

These TFs regulate gene transcription by binding to the conserved W-box in the promoter region of target genes. WRKY TFs are categorized into three groups based on the number of WRKY domains (two WRKY domains in Group I and one in Group II and III) and the structure of the zinc fingers ($C_2H_2$ in Group I and II, and $C_2HC$ in Group III proteins). Group II is further divided into five subgroups (IIa-IIe) based on the phylogeny of the WRKY domain (*Chanwala et al., 2020*; *Rushton et al., 2010*). WRKY TFs can function individually or in combination with other regulators to activate or suppress transcription (*Jiang et al., 2017*).

The first member of the WRKY gene family, *SPF1*, was cloned from sweet potato in 1994 by *Ishiguro & Nakamura (1994)*. Subsequently, genes encoding WRKY proteins have been discovered in numerous plants, such as 71 members of *Arabidopsis thaliana* (*Song & Gao, 2014*) 78 members of garlic (*Allium sativum* L.) (*Yang et al., 2021*), 174 members of soybean (*Glycine Max*) (*Yang et al., 2017*), and 104 members in rice (*Oryza sativa*) (*Jeyasri et al., 2021*). The WRKY gene family exhibits wide distribution in plants and plays a crucial role in plant growth and development. For instance, wild strawberry (*FvWRKY4, -46,* and *-48*) in group IIc is involved in fruit development and ripening in *F. vesca* (*Zhou et al., 2016*). Overexpression of *CpWRKY71* in *Arabidopsis* leads to early flowering and premature senescence phenotypes (*Wang, Li & Yu, 2011*. Several studies have confirmed that WRKY TFs are involved in seed size (*Gu et al., 2017*), growth types (*Yang et al., 2016*), and hormone signaling pathways (*Sun et al., 2022*). Additionally, WRKY proteins have been extensively studied and reported to abiotic and biotic stresses. Research in bananas found that the *WRKY71* gene was induced by low temperature, salt, abscisic acid (ABA), $H_2O_2$ and ethylene (*Shekhawat, Ganapathi & Srinivas, 2011*). Transcriptome analysis in *Populus* revealed that out of 100 WRKY genes, 61 were induced by salicylic acid (SA), methyl jasmonate (MeJA), injury, cold, and salt stresses (*Wang et al., 2020*). Furthermore, *Brassica napus* L.'s *WRKY71* family genes were up-regulated by 4 °C treatment, suggesting their positive regulatory role in cold tolerance by regulating downstream target genes (*Yuan et al., 2018*). Overexpression of *MfWRKY17* can increase the water retention capacity of *Arabidopsis*, maintain chlorophyll content, regulate ABA biosynthesis, and influence the transcription level of related genes, enhancing drought resistance (*Wang et al., 2021*). Plant responses to environmental stress are regulated by complex signaling pathways and networks coordinated by TFs.

Yam (*Dioscorea opposita* Thunb., *D.opposita*), belonging to the genus *Dioscorea* in the Dioscoreaceae family, is a monocotyledonous vine. As one of the top 10 most important edible tuber and root plants, various yam species have been domesticated and extensively cultivated for nutritional and medicinal purposes (*Shan et al., 2020*; *Cao et al., 2021*). The tuber serves as the main product organ of yam and undergoes a complex growth and development process (*Yin et al., 2022*). Throughout cultivation, yam is vulnerable to various biological and abiotic stresses that will affect tuber yield and quality. Therefore, understanding the tuber development mechanism is crucial for improving yam yield. With the advancement of high-throughput sequencing technology, the WRKY gene family has been identified in various plants, including *Arabidopsis thalianathe* (*Eulgem et al., 2000*), *Oryza sativa* (*Ross, Liu & Shen, 2007*), *Triticum aestivum* (*Ye et al., 2021*), *Solanum*

*tuberosum* (*Villano et al., 2020*), and tomato (*Huang & Liu, 2013*). However, information on WRKY characteristics in yam remains limited. In this study, we isolated and identified 25 WRKY proteins in yam from the transcriptome database and subjected them to phylogenetic analysis, gene structure analysis, and conserved domain alignment. Additionally, we conducted a comprehensive analysis of *WRKY* genes in yam, their expression patterns at different growth and development stages, and their responses to abiotic treatments. Furthermore, we assessed the impact of overexpressing *DoWRKY71* on improved cold and ABA tolerance by analyzing phenotypic and physiological changes, as well as the expression of stress-response genes in tobacco. This study lays a theoretical foundation for further research on the role of *WRKY* regulatory networks in yam growth, development and stress-related processes.

## MATERIALS AND METHODS

### Identification of WRKY gene family in *D. opposita*

Due to the lack of genomic data of this species, the WRKY gene family sequence of yam was identified by the full-length transcriptome data of yam tuber at different developmental stages in our laboratory (Nuohezhiyuan Company, Beijing in 2020). Hidden Markov model (HMM) profiles based on the *Oryza sativa* WRKY (PFAMID: PF03106.15) protein from the PFAM database (http://pfam.xfam.org/) were used (*Ross, Liu & Shen, 2007*) (Table S1). BLASTp analysis of the *D.opposita* transcriptome database was performed using HMMER 3.0., setting the cut-off E-value as 0.1. The presence of WRKY motif was confirmed using the online program PFAM (http://pfam.xfam.org/search), Conserved Domains DataBase Search (https://www.ncbi.nlm.nih.gov/Structure/cdd/wrpsb.cgi) and SMART tool (http://smart.embl-heidelberg.de/). The open reading frame (ORF) of the candidate sequences with complete WRKY transcription factors was predicted from the search results using ORF Finder (https://www.ncbi.nlm.nih.gov/gorf/gorf.html) (Portions of this text were previously published as part of a preprint (https://www.researchsquare.com/article/rs-2164173/v1)) (*Xing et al., 2022*).

### Sequence analyses and conserved motif detection of *D. opposita*

Biophysical properties of WRKY members, such as peptide length, molecular weight (MW), and isoelectric point (pI) were predicted using the online ExPasy program (http://www.expasy.org/tools/) (*Wilkins et al., 1999*). Protein subcellular localization prediction was done using an online website (https://wolfpsort.hgc.jp/). The online software MEME (https://meme-suite.org/meme/) was employed to identify conserved motifs among all *DoWRKY* genes and annotated with InterPro Scan. The parameters used in this study were as follows: 0 or 1 occurrence per sequence; maximum number of motifs: 10; and other optional parameters were set to default (*Bailey et al., 2015*).

### Multiple sequence alignment and phylogenetic analysis

All 25 DoWRKYs proteins with WRKY domains and 44 selected OsWRKYs proteins were used to generate multiple protein sequence alignments using Clustal Omega with default settings. A phylogenetic tree was constructed with the maximum likelihood method with

MEGAX software (*Tamura et al., 2011*). Branch support for the tree topology was estimated through bootstrap analysis with 1,000 replicates. The online ITOL (http://itol.embl.de/help.cgi) tool was used to visualize the phylogenetic tree.

## Plant materials and treatments

Tissue culture seedlings of *D. opposita* (*Dioscorea opposita* Thunb., Dahechangyu, *D. opposita*) was used as the experimental material. The seedlings were cultured in an artificial climatic chamber at 25 °C with a 16/8 h photoperiod (day/night). For stress treatments, 6-week-old seedlings were exposed to different conditions: ABA (100 μmol/L), cold (4 °C), MeJA (100 μmol/L), and cultured for a total of 24 h. Leaves were collected at 0, 1, 6, 12 and 24 h after treatment. For different developmental stages, yam tubers of each cultivar were randomly sampled in the morning at six different stages: after 90, 105, 120, 135, 150 and 165 days. All samples were immediately frozen in liquid nitrogen and stored at −80 °C for subsequent total RNA extraction and gene expression analysis. Three biological replicates were used for each sample.

## RNA extraction and quantitative real-time PCR

Total RNA was extracted using RNA extraction kit (9769; TaKaRa, Beijing, China) according to the manufacturer's instructions. First-strand cDNA synthesis was performed using TransScript One-Step gDNA Removal and cDNA Synthesis SuperMix (RR047A; TaKaRa, Beijing, China). The quality of the RNA samples were tested using NanoDrop2000c (Thermo Fisher Scientific, Waltham, MA, USA) and gel electrophoresis. Quantitative real-time PCR (qRT-PCR) was carried out using TB Green Premix Ex Taq II (6210A; TaKaRa, Beijing, China) on an FTC-3000P system (Funglyn Biotech, Toronto, Canada). Design the primer used Primer Premier 5.0 with the primers listed in Table S2. The PCR conditions consisted of denaturation at 95 °C for 30 s, followed by 40 cycles of 5 s at 95 °C, and 30 s at 60 °C, and a final step at 4 °C. All samples were tested with three technical replicates and three independent biological replicates. The relative expression level was calculated using the $2^{-\Delta\Delta CT}$ method with yam ribosomal RNA (18S) as the internal control (internal reference genes screened by our laboratory) (*Shao et al., 2021*).

## Subcellular localization analysis

After screening the transcriptome sequencing results of yam and subsequent studies, the *DoWRKY71* gene was found to be responsive to tuber expansion, cold, ABA, and MeJA stresses, and was selected for further research. Subsequently, its full length was cloned, and it was named *DoWRKY71* (Genbank number: OP380252) after comparison with NCBI. The ORF of *DoWRKY71* was amplified using the specific primers, *DoWRKY71*-SF/SR, which contained restriction enzyme cutting sites (Table S2). Subsequently, the amplified ORF was inserted into the CaMV35S::GFP vector. Following sequence confirmation, the CaMV35S::GFP- *DoWRKY71* recombinant plasmid and CaMV35S::GFP (control) plasmids were transferred into *Agrobacterium tumefaciens* LBA4404 (Weidi Biotechnology, Shanghai, China). The empty CaMV35S::GFP vector was used as the

control. Two types of *Agrobacterium tumefaciens* were then used to infect *Nicotiana benthamiana* leaves. After 2–3 days, infected parts of leaves were cut and used for fluorescent signal detection using a laser scanning confocal microscope (C2-ER; Nikon, Tokyo, Japan) (*Aung et al., 2017*).

## Generation and identification of transgenic plants

The encoding sequence of *DoWRKY71*, along with the restriction sites of *Kpn*I and *BamH*I (specific primers, *DoWRKY71*-ZF/ZR), was cloned into pPZP221 to construct the expression vector pPZP221-*DoWRKY71*. The *Agrobacterium tumefaciens* strain EHA101 (Weidi Biotechnology, Shanghai, China) was used to transform the wildtype (WT) tobacco *via* the freeze-thaw method. The plants were selected on MS medium supplemented with cefotaxim (500 mg/L) and gentamicin (50 mg/L). DNA and RNA from the leaves of transgenic and WT tobacco were detected using Plant Kit (DNA: DP305; TIANGEN, Beijing, China; RNA: 9769; TaKaRa, Beijing, China). Three positive transgenic lines (T2, T3 and T4) were selected for further gene-functional verification.

## Treatment and functional analysis of overexpressed *DoWRKY71* in tobacco

The 4-week-old transgenic and WT tobacco plants were transplanted into plug trays and allowed to grow for 2 weeks. Transgenic and WT tobacco plants with similar growth activity were selected and treated with ABA (100 μmol/L) and 4 °C, and their leaves were selected at 0, 1, 6, 12 and 24 h after treatment. Collection and storage methods are the same as plant materials and treatments. WT tobacco was used as control to determine various physiological indicators, including chlorophyll content, soluble protein and malondialdehyde (MDA) content, peroxidase (POD), superoxide dismutase (SOD) (*Li, 2000*). The content of gibberellin (GA) and ABA were measured using the kit (Shanghai Best Biotechnology Co., Ltd., Shanghai, China). The production rate of superoxide anion radical ($O_2\cdot$-) and the content of hydrogen peroxide ($H_2O_2$) were detected by staining of nitroblue tetrazolium (NBT) and diaminobenzine (DAB) (*Yang et al., 2018*).

The experiments were repeated at least three times, with three biological replicates for each sample. Stomatal openings of transgenic and WT tobacco leaves under ABA treatment were observed and photographed with an electron microscope (C2-ER; Nikon, Tokyo, Japan). The index of pore size was calculated using Digimizer software. The relative expression levels of *DoWRKY71* gene in transgenic tobacco were determined at different time points during 4 °C and ABA treatments.

## Statistical analysis

All data were based on three replicates. Excel 2019 was used for data processing and mapping. SPSS 25 (IBM, Armonk, NY, USA) was used for statistical analysis. The gene expression and physiological indicator data were tested with $p < 0.05$ or $p < 0.01$ significance level.

## RESULTS

### Identification and sequence analysis of *D. opposita*

After manually removing redundant entries through screening (manually deleting the genes without ORF/WRKY motif) and validation of the search results, a number of 25 WRKY family genes were characterized from *D. opposita*. 25 proteins were named according to NCBI database comparison and their basic characteristics are shown Table 1. The coding sequence lengths of DoWRKY proteins varied from 191 to 718 amino acids, from 21.56 to 78.02 kDa in molecular weight, and from 4.88 to 9.93 in their isoelectric points. Different amino acids have different isoelectric points due to their different structures. The isoelectric point of acidic amino acid solution must be less than seven, and the isoelectric point of basic amino acid must be greater than seven. Of the 25 proteins we isolated, 12 were basic and 13 were acidic. Protein molecular weight is mainly used to determine the size of proteins, and can be used to measure the complexity of proteins. Members of the same gene family have differences in protein size and complexity. The subcellular localization of the DoWRKY proteins was predicted by using WOLF PSORT, which indicated that most of the DoWRKY proteins were likely located in the nucleus, with only a few in the plasmolemma (*Chang et al., 2022*; *Li et al., 2024*).

### Phylogenetic analysis and multiple sequence alignment of DoWRKY proteins

To investigate the evolutionary relationships and classification of the DoWRKY and *Oryza* WRKY (OsWRKY) proteins, a phylogenetic tree was constructed with the 25 DoWRKY proteins and OsWRKY proteins (Fig. 1). The results indicated that the 25 DoWRKY proteins were divided into three categories (I–III) according to the groups in *Oryza*. Group I contained the highest WRKY members with 7 DoWRKYs, containing one $C_2H_2$ zinc-finger motif and two WRKY domains. A total of 15 DoWRKY proteins included one WRKY domain and one $C_2H_2$ zinc-binding motif, which were considered to be Group II. Among them, Group lla contains 5 DoWRKY proteins, Group IIb contains none, Group IIc contains three DoWRKY proteins, Group IId contains five DoWRKY proteins, and Group IIe contains two DoWRKY proteins. There are three DoWRKY proteins classified as Group III, containing a single WRKY domain and a $C_2HC$ zinc-binding motif Table 1.

Based on previous studies, we randomly selected the representative sequence of *Oryza* for comparison with yam. In addition, conserved amino acids in the WRKY domain analysis *via* multiple sequence alignments of the DoWRKY proteins were performed. Among the 25 DoWRKY proteins in the yam, 24 proteins have the "WRKYGQK", the core dominant terminal. While only one protein (DoWRKY50) has single amino acid substitutions in the WRKYGQK domain and "WRKYGKK" domain. The 22 DoWRKY proteins had $C_2H_2$ zinc-finger, and 3 DoWRKY proteins (DoWRKY70, DoWRKY38 and DoWRKY46) had $C_2HC$ zinc-finger at the N-terminal (Figs. 2 and 3).

The classification of DoWRKYs confirmed the diversity of their protein structures, suggesting that different subfamily members may have different regulatory functions, and

**Table 1 List of all WRKY transcription factor identified in *D. opposita*.**

| Gene ID | Gene name | ORF (aa) | PI | MW (kDa) | Subcellular localization | Family group | Zinc finger | Domain pattern | WRKY domain |
|---|---|---|---|---|---|---|---|---|---|
| transcript_HQ_D_transcript5735 | DoWRKY2a | 718 | 5.78 | 78.02 | Nuclear | I | $C_2H_2$ | CX4-CX23-HXH | WRKYGQK/ WRKYGQK |
| transcript_HQ_D_transcript6292 | DoWRKY2b | 709 | 5.56 | 76.91 | Nuclear | I | $C_2H_2$ | CX4-CX23-HXH | WRKYGQK/ WRKYGQK |
| transcript_HQ_D_transcript14202 | DoWRKY20 | 650 | 6.16 | 70.43 | Nuclear | I | $C_2H_2$ | CX4-CX23-HXH | WRKYGQK/ WRKYGQK |
| transcript_HQ_D_transcript19865 | DoWRKY4 | 528 | 6.64 | 57.68 | Nuclear | I | $C_2H_2$ | CX4-CX23-HXH | WRKYGQK/ WRKYGQK |
| transcript_HQ_D_transcript20189 | DoWRKY44 | 283 | 8.88 | 31.51 | Nuclear | I | $C_2H_2$ | CX4-CX23-HXH | WRKYGQK/ WRKYGQK |
| transcript_HQ_D_transcript20271 | DoWRKY32 | 529 | 5.24 | 56.35 | plasmolemma | I | $C_2H_2$ | CX4-CX23-HXH | WRKYGQK/ WRKYGQK |
| transcript_HQ_D_transcript23159 | DoWRKY33 | 517 | 6.93 | 57.13 | Nuclear | I | $C_2H_2$ | CX4-CX23-HXH | WRKYGQK/ WRKYGQK |
| transcript_HQ_D_transcript23749 | DoWRKY21 | 357 | 9.74 | 39.99 | Nuclear | IId | $C_2H_2$ | CX5-CX23-HXH | WRKYGQK |
| transcript_HQ_D_transcript28076 | DoWRKY22 | 345 | 5.6 | 37.37 | Nuclear | IIe | $C_2H_2$ | CX5-CX23-HXH | WRKYGQK |
| transcript_HQ_D_transcript29385 | DoWRKY40a | 306 | 7.66 | 34.2 | Nuclear | IIa | $C_2H_2$ | CX5-CX23-HXH | WRKYGQK |
| transcript_HQ_D_transcript29497 | DoWRKY71a | 277 | 9.04 | 31.13 | Nuclear | IIa | $C_2H_2$ | CX5-CX23-HXH | WRKYGQK |
| transcript_HQ_D_transcript29812 | DoWRKY40b | 308 | 8.2 | 34.51 | Nuclear | IIa | $C_2H_2$ | CX5-CX23-HXH | WRKYGQK |
| transcript_HQ_D_transcript30455 | DoWRKY40c | 307 | 7.06 | 34.29 | Nuclear | IIa | $C_2H_2$ | CX5-CX23-HXH | WRKYGQK |
| transcript_HQ_D_transcript30533 | DoWRKY39 | 241 | 9.93 | 27.44 | Nuclear | IId | $C_2H_2$ | CX5-CX23-HXH | WRKYGQK |
| transcript_HQ_D_transcript30703 | DoWRKY71b | 296 | 8.87 | 33.09 | Nuclear | IIa | $C_2H_2$ | CX5-CX23-HXH | WRKYGQK |
| transcript_HQ_D_transcript31055 | DoWRKY70 | 280 | 5.17 | 31.59 | Nuclear | III | C2HC | CX7-CX23-HXC | WRKYGQK |
| transcript_HQ_D_transcript31596 | DoWRKY71 | 320 | 6.36 | 35.82 | Nuclear | IIc | $C_2H_2$ | CX4-CX23-HXH | WRKYGQK |
| transcript_HQ_D_transcript31770 | DoWRKY38 | 311 | 4.98 | 35.07 | Nuclear | III | C2HC | CX6-CX24-HXC | WRKYGQK |
| transcript_HQ_D_transcript31912 | DoWRKY51a | 314 | 9.67 | 33.74 | Nuclear | IId | $C_2H_2$ | CX5-CX23-HXH | WRKYGQK |
| transcript_HQ_D_transcript32563 | DoWRKY51b | 309 | 9.9 | 33.38 | Nuclear | IId | $C_2H_2$ | CX5-CX23-HXH | WRKYGQK |
| transcript_HQ_D_transcript32736 | DoWRKY69 | 264 | 4.88 | 28.04 | Nuclear | IIe | $C_2H_2$ | CX5-CX23-HXH | WRKYGQK |
| transcript_HQ_D_transcript33996 | DoWRKY46 | 231 | 6.49 | 26.03 | Nuclear | III | C2HC | CX7-CX23-HXC | WRKYGQK |
| transcript_HQ_D_transcript34355 | DoWRKY51c | 264 | 9.93 | 28.69 | Nuclear | IId | $C_2H_2$ | CX5-CX23-HXH | WRKYGQK |

*(Continued)*

| Gene ID | Gene name | ORF (aa) | PI | MW (kDa) | Subcellular localization | Family group | Zinc finger | Domain pattern | WRKY domain |
|---------|-----------|----------|-----|----------|--------------------------|--------------|-------------|----------------|-------------|
| transcript_HQ_D_transcript35143 | DoWRKY50 | 191 | 6.51 | 21.56 | Nuclear | IIc | $C_2H_2$ | CX4-CX23-HXH | WRKYGKK |
| transcript_HQ_D_transcript35799 | DoWRKY12 | 199 | 8.55 | 23.06 | Nuclear | IIc | $C_2H_2$ | CX4-CX23-HXH | WRKYGQK |

**Note:**
The physical properties of WRKY members, such as ORF length, isoelectric point (pI), molecular weight (MW), subcellular localization, zinc-finger type and WRKY domain.

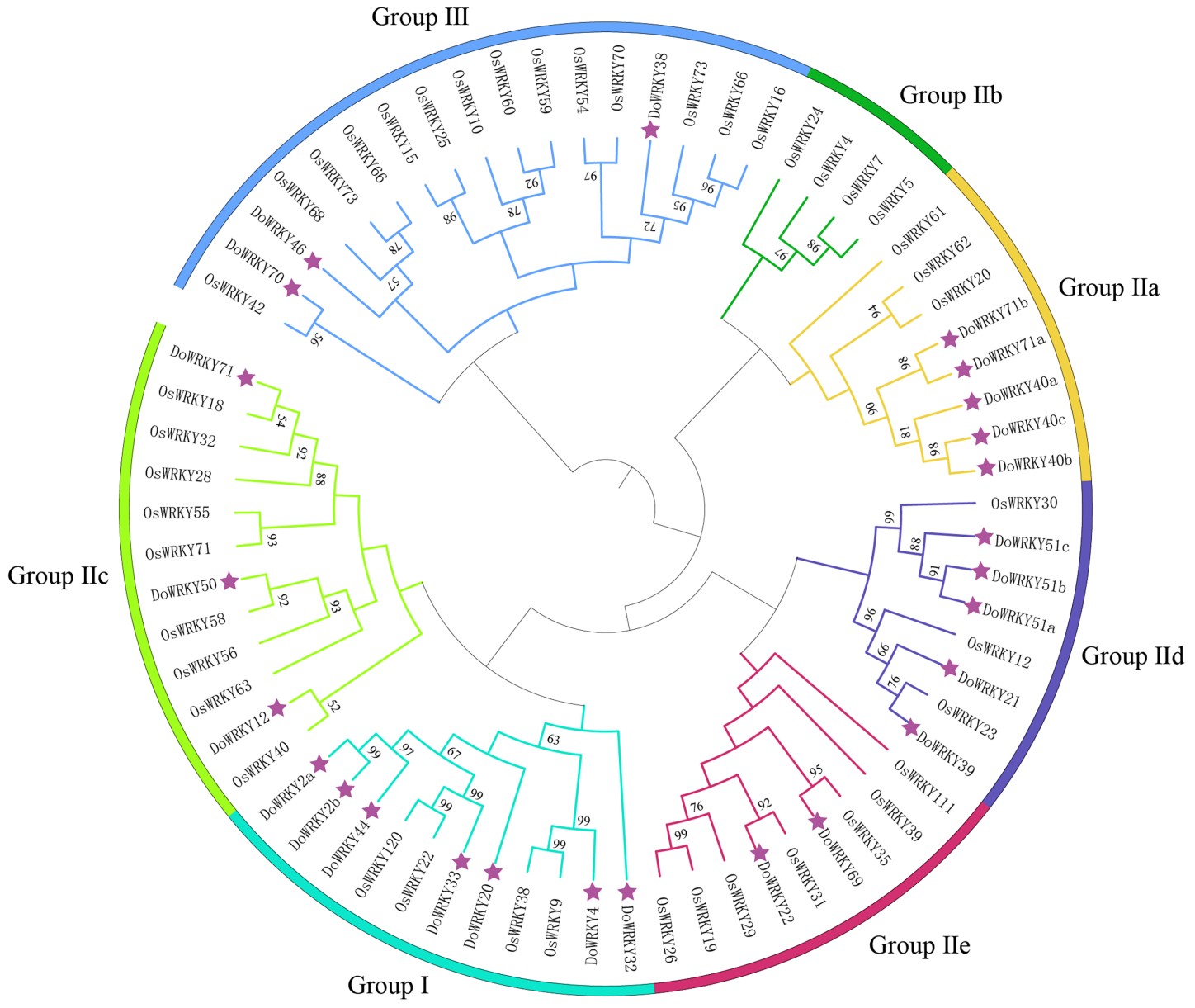

**Figure 1 Phylogenetic tree analysis of the WRKY proteins in *D. opposita* and *Oryza*.** The clusters were designated as group I–III. Different color branches and strips are used to distinguish different subgroups.

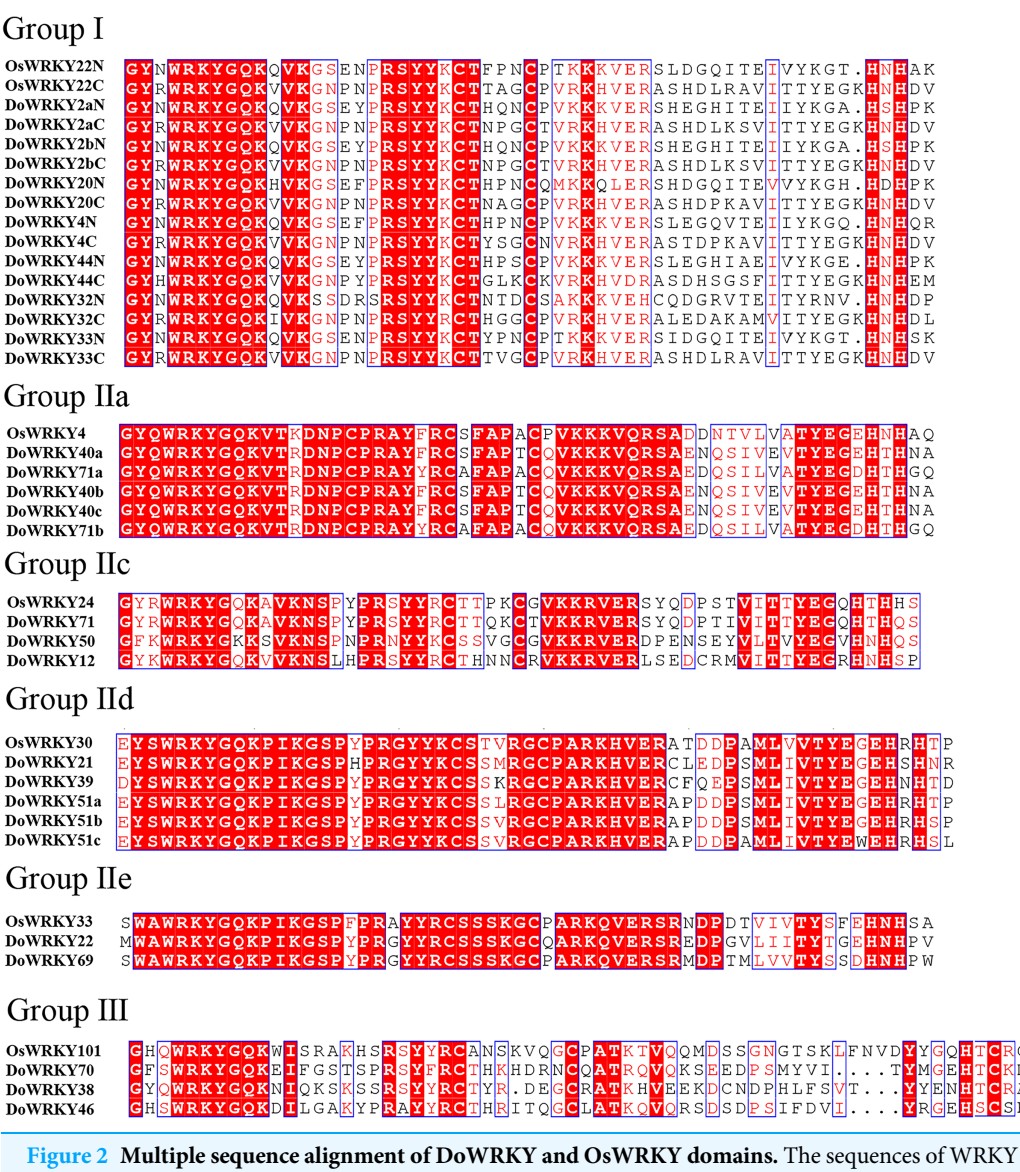

Figure 2 **Multiple sequence alignment of DoWRKY and OsWRKY domains.** The sequences of WRKY variants were aligned. The conserved WRKYGQK and zinc-finger residues were marked in red.

that orthologous genes are likely to have similar functions. These results revealed that WRKY proteins existed in various plant species.

## Conserved motifs and sequence alignment of DoWRKY proteins

To further analyze the characteristics of DoWRKYs, the MEME online tool was used to predict the potential conserved motifs. A total of 10 motifs were identified, with lengths ranging between 16 and 50 amino acids (Fig. 4). Motif 1 represents the WRKY domain, and Motif 2 represents the zinc finger motif. These two domains were the two highly conserved motifs contained present in every WRKY member. DoWRKY40a, DoWRKY40b and DoWRKY40c contained the highest number of conservative motifs, including six. DoWRKY38, DoWRKY69, DoWRKY46 and DoWRKY50 contained the

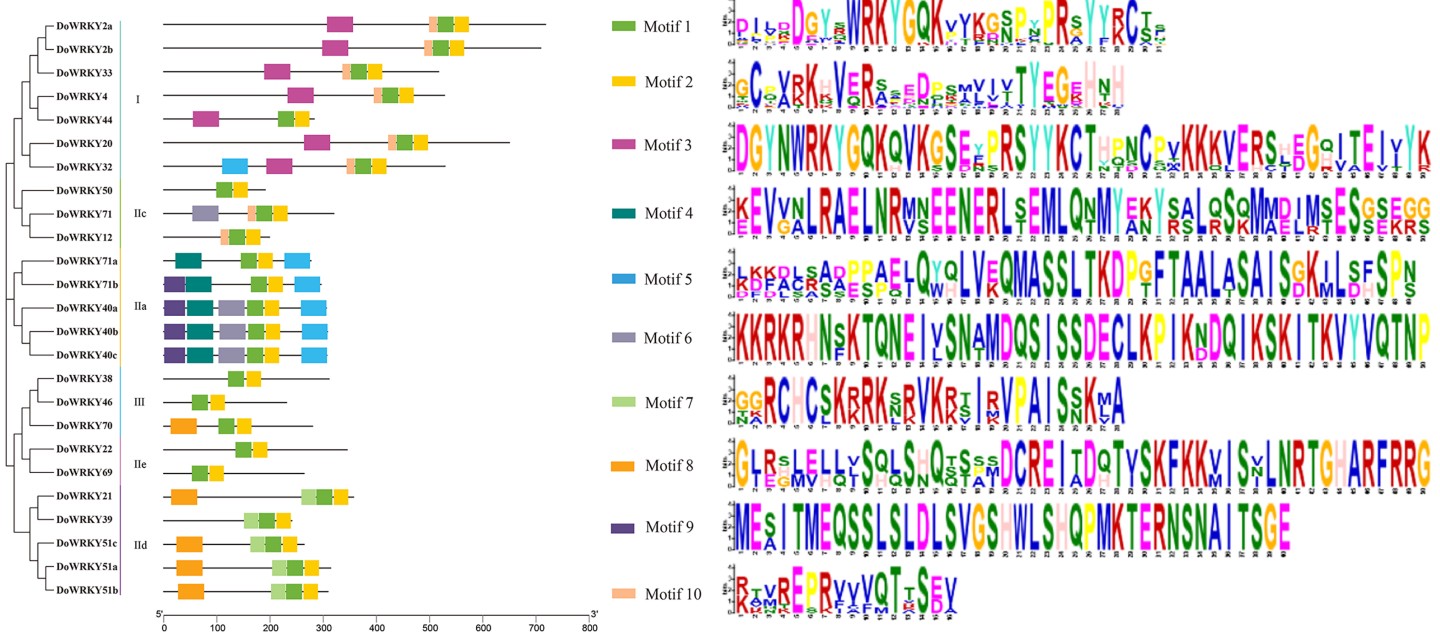

**Figure 3 Multiple sequence alignment of WRKY proteins in *D. opposita*.** The sequences of WRKY variants were aligned. The conserved WRKYGQK and zinc-finger residues were marked in red.

**Figure 4 Conserved motifs analysis of the DoWRKY proteins.** The different colored boxes represent different motifs and their position in each DoWRKY sequence. Each motif is indicated by a colored box in the legend at the right.

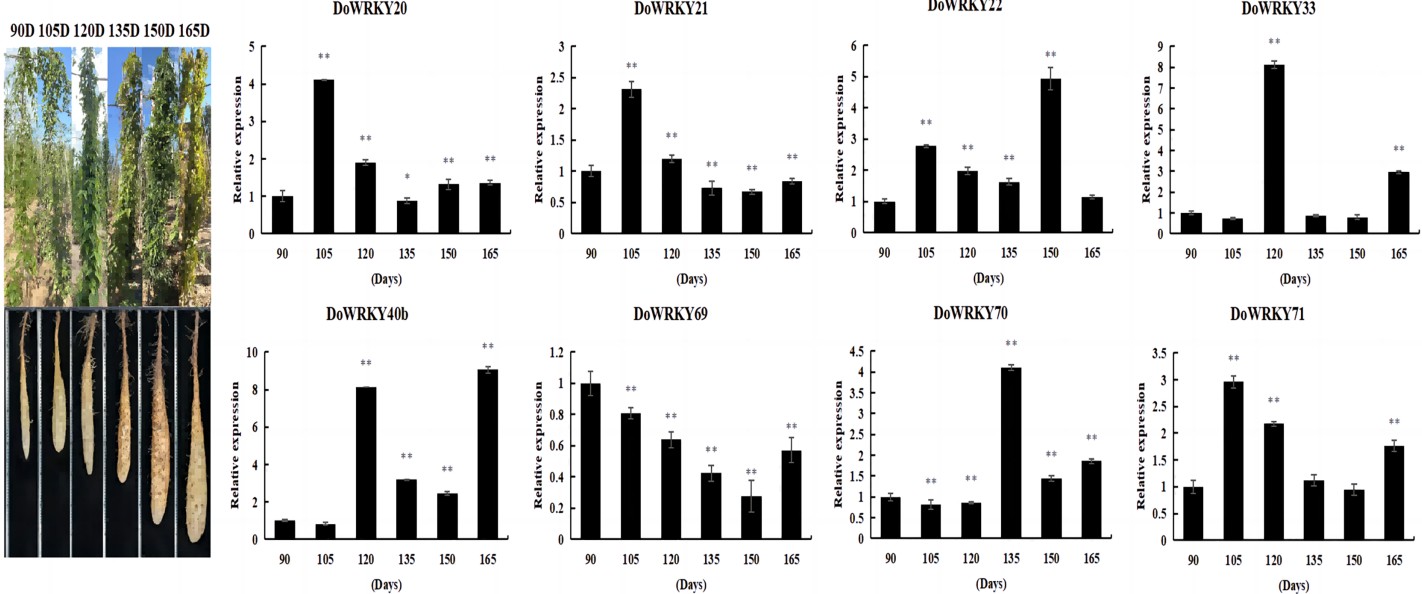

**Figure 5 Expression analysis of the *WRKY* genes in tuber different growth stages of *D. opposita*.** Data are presented as mean ± SD (*n* = 3). * and ** indicate a significant difference at *P* < 0.05 and *P* < 0.01 compared with 90 d, respectively.

fewest number of conservative motifs, only two. The clustered DoWRKY pairs, such as DoWRKY2a/2b, DoWRKY51a/51b, and DoWRKY40a/40c, displayed a highly similar motif distribution. The conserved motifs of the same subgroup of DoWRKY proteins are similar, but discrepant between groups. This conserved domain analysis reflected the similarity of amino acid sequences within groups and the difference of amino acid sequences between groups. This also suggested that the phylogenetic classification was reliable.

## Expression analysis of *DoWRKY*s in different developmental stages of yam tuber

By comprehensively considering the protein family classification and subcellular localization, we selected eight genes from six groups to analyze the expression level of yam under different growth and development stages. The expression of *DoWRKY40b* was down-regulated at 105 days and up-regulated at other periods, reaching the highest value at 165 days. *DoWRKY69* was down-regulated in the whole process. *DoWRKY33* reached its peak expression at 105 d, which is 8.11 times higher than at 90 days. Three genes *DoWRKY20*, *DoWRKY21* and *DoWRKY71*, showed the same trend: up-regulation expressed in the earlier stage and down-regulation expression in the mature stage (Fig. 5).

## *DoWRKYs* gene expression following abiotic stress by qRT-PCR

To further investigate the role of the 8 *DoWRKYs* in abiotic stress responses, we subjected them to various treatments including cold, ABA and MeJA.

Following cold treatment (Fig. 6A), the expression levels of two genes (*DoWRKY40b* and *DoWRKY71*) increased by more than twofold, among which the expression of *DoWRKY40b* was 6.38 times and *DoWRKY71* was 11.53 times higher than that of the

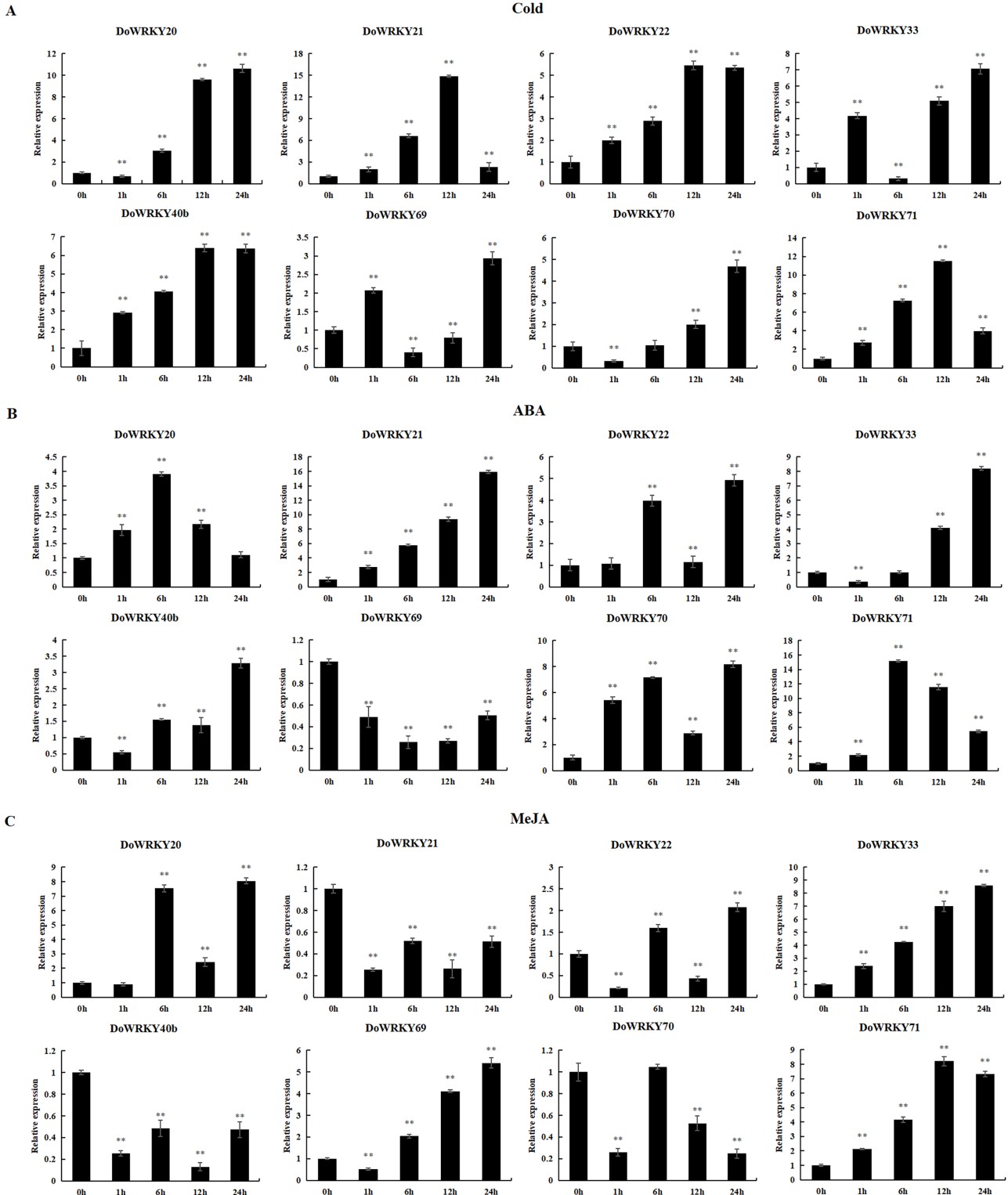

**Figure 6  qRT-PCR validation of *WRKY* genes in the response to cold, ABA, MeJA treatment.** (A) Cold treatment; (B) ABA treatment; (C) MeJA treatment. Data are presented as mean ± SD (*n* = 3). ** indicates a significant difference at *P* < 0.01 compared with 0 h, respectively.

control. The expression of two genes (*DoWRKY33* and *DoWRKY69*) increased significantly at 1 h, decreased at 6 h, and then their expression significantly increased and peaked at 24 h. The expression levels of *DoWRKY22* and *DoWRKY40b* genes showed an overall upward trend, and became stable at 12 h. The expression levels of *DoWRKY21* and *DoWRKY22* genes showed an overall upward trend, reaching the highest level at 12 h and becoming stable.

After ABA treatment, the expressions of two genes (*DoWRKY20* and *DoWRKY71*) first increased and then decreased (Fig. 6B). The expression levels of *DoWRKY21* increased significantly over the control. The expression levels of *DoWRKY33* and *DoWRKY70* were about eight times higher than that of the control 24 h after spraying treatment. The expression levels of *DoWRKY69* decreased significantly compared to the control.

For the case of MeJA stress (Fig. 6C), the expression of *DoWRKY21* and *DoWRKY40b* showed a downward trend. *DoWRKY33* was up-regulated during the whole process, and the expression at hour 24 was 8.21 times that of 0 h. The expression level of the three genes *DoWRKY20*, *DoWRKY33* and *DoWRKY69* reached the highest value at 24 h.

## Subcellular localization of DoWRKY71

To furnish additional evidence for the plausible involvement of DoWRKY71 in transcriptional regulation, we constructed a DoWRKY71-GFP fusion vector alongside a GFP fusion expression empty vector for use as a control. These vectors were driven by the CaMV35S promoter and were subsequently introduced into the epidermis of tobacco leaves through the Agrobacterium-mediated transient expression method. Subsequently, they were then visualized under a confocal microscope. The GFP fluorescence emanating from the CaMV35S-GFP-DoWRKY71 exhibited a robust signal solely within the nucleus, whereas the CaMV35S-GFP vector displayed a more even distribution throughout the tobacco cells (Fig. 7).

## Identification and regeneration of transgenic tobacco with *DoWRKY71* gene

To examine the cold stress tolerance of *DoWRKY71* in transgenic tobacco, we employed the recombinant vector pPZP221-*DoWRKY71*. Transgenic lines were meticulously selected for PCR and RT-PCR detection, which indicated the successful integration of the *DoWRKY71* gene into the tobacco genomic DNA and its expression at the transcriptional level. Subsequently, three independent transgenic lines overexpressing *DoWRKY71*(T2-T4) were acquired through resistance selection to facilitate subsequent experiments (Fig. 8).

## Physiological characteristics of overexpressed *DoWRKY71* tobacco under cold stress

During our investigation of the cold stress tolerance of *DoWRKY71* in transgenic tobacco, we observed that the growth of transgenic tobacco under control conditions surpassed that of the wild-type tobacco, and the cold tolerance of transgenic tobacco exhibited notable enhancement (Fig. 9A). Furthermore, the chlorophyll content of the transgenic plants was

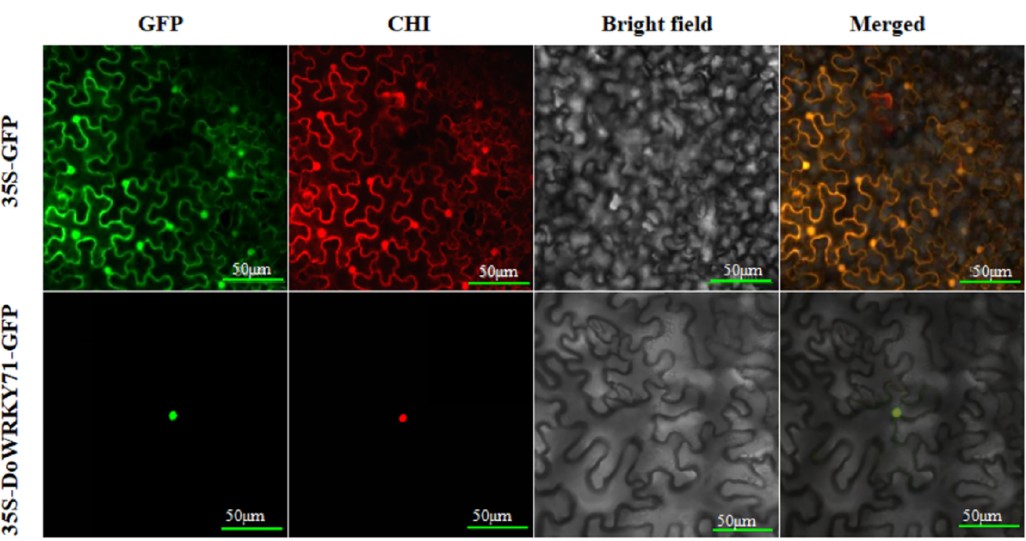

**Figure 7 DoWRKY71 subcellular localization.** The excitation wavelength of confocal laser microscope was set as: GFP 488 nm; Chloroplast 561 nm; Bright field:DIC; Merge: superimposed field; Bar = 50 μm.

**Figure 8 Identification and regeneration of transgenic tobacco with *DoWRKY71* gene.** (A) Schematic diagram of recombinant vector pPZP221-DoWRKY71. (B) PCR detection of DoWRKY71 gene tobacco. (C) RT-PCR detection of DoWRKY71 gene tobacco. M1 and M2 = DL2000 DNA marker; + = plasmid pPZP221-*DoWRKY71* as positive control; − = water as negative control; WT = wild type; 1–9 = different transgenic lines.

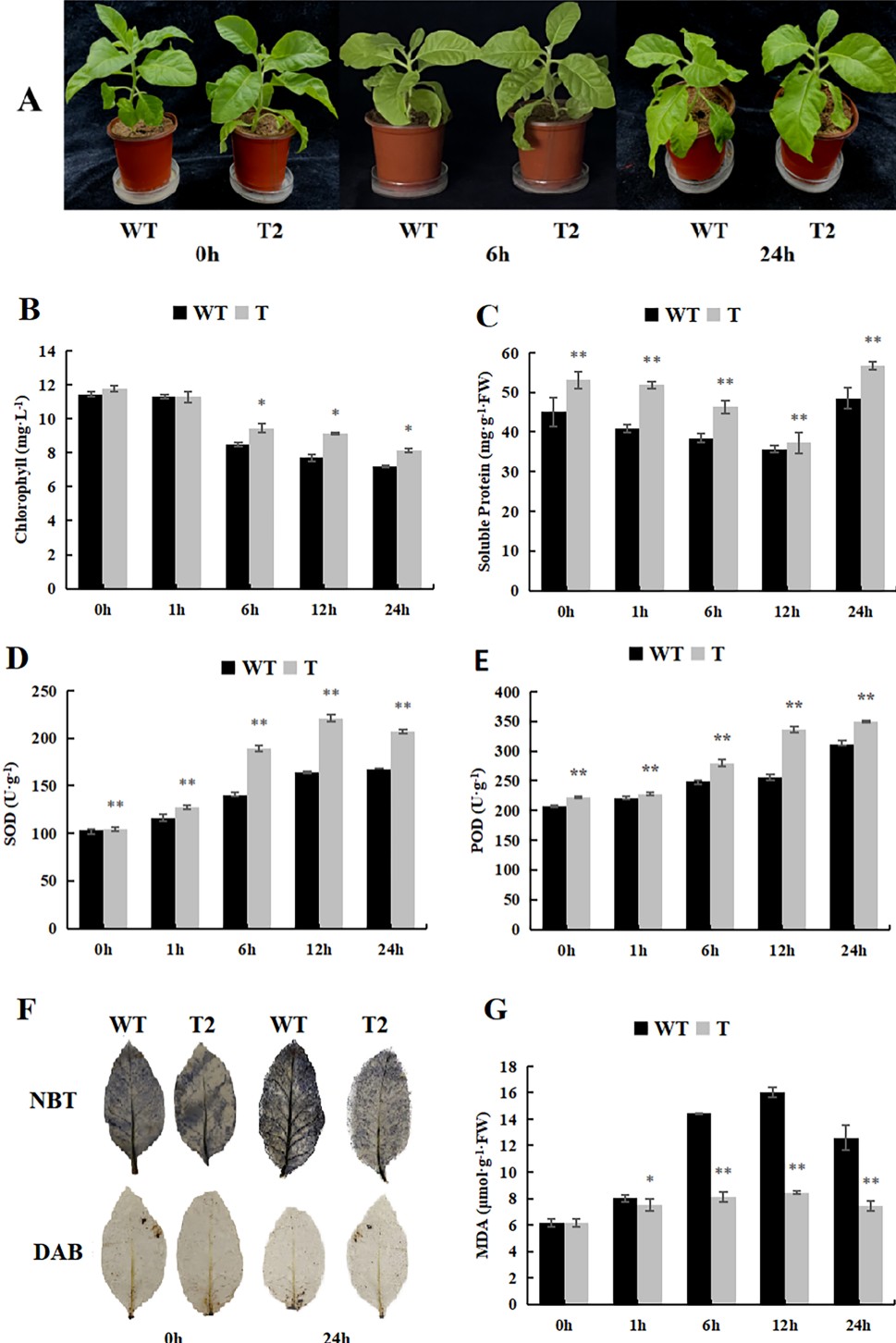

**Figure 9 Growth, physiological and biochemical indexes of WT and transgenic tobacco overexpressing yam *DoWRKY71* in cold temperature.** (A) Plant growth; (B) chlorophyll content; (C) soluble protein; (D) SOD activity; (E) POD activity; (F) DAB and NBT staining of transgenic plant leaves; (G) MDA content in the transgenic lines and WT plants under cold conditions. 0 h: (CK), 6 h: 6 h after treatment, 24 h : 24 h after treatment. WT, wild-type tobacco; T2, transgenic tobacco. Data are presented as mean ± SD (*n* = 3). * and ** indicate a significant difference at *P* < 0.05 and *P* < 0.01 compared with WT, respectively.

significantly higher than that of the wild-type plants after 6 h of treatment (Fig. 9B). Notably, the soluble protein content initially declined, reaching its peak 24 h after treatment began, and the transgenic plants displayed significantly higher levels in comparison to the wild-type plants (Fig. 9C). Moreover, the activities of SOD and POD demonstrated an upward trend, with the transgenic plants exhibiting significantly higher levels than the wild-type plants (Figs. 9D and 9E).

Low temperature typically induces membranous peroxidation and overproduction of reactive oxygen species, eventually leading to oxidative stress. NBT and DAB staining were performed on tobacco subjected to cold treatment (Fig. 9F), and while staining was evident in both control wild-type and transgenic plants, the $O_2 \cdot -$ and $H_2O_2$ levels were lower than those observed in the wild-type plants. Additionally, MDA content in both transgenic and wild-type plants increased with time during low-temperature stress, but experienced a slight decrease after 24 h. Notably, the MDA content accumulated by transgenic lines in control and experimental groups was significantly lower than that of the wild-type plants (Fig. 9G). These findings suggest that overexpression of *DoWRKY71* regulates reactive oxygen species homeostasis and reduces oxidative damage in tobacco.

## Physiological characteristics of overexpressed *DoWRKY71* tobacco under ABA treatment

As the spraying time extended, the soluble protein content initially declined and then rose. In both the control and treatment groups, the transgenic plants exhibited significantly higher soluble protein content than that observed in the wild-type tobacco (Fig. 10A). Furthermore, the trends in SOD and POD activity were essentially similar to those observed under cold treatment, with POD activity significantly higher than that observed in the wild-type tobacco (Figs. 10B and 10C). As for GA content, it displayed an overall upward trend, peaking at 12 h, whereas ABA content initially increased and then decreased. The GA content in the transgenic lines, both in the control and experimental groups, was significantly lower than that observed in the wild-type plants (Fig. 10D), while ABA content was significantly higher than that in the wild-type plants (Fig. 10E).

Microscopic observation (100×) of the stomata in *DoWRKY71* transgenic plants and wild-type plants under ABA treatment revealed that, under normal conditions, the stomatal pore size index of T3 and T4 was significantly larger than that observed in the wild-type plants (Fig. 11A). However, upon treatment with 100 μmol/L ABA spray, the stomatal index of *DoWRKY71* transgenic lines was significantly smaller than that observed in the wild-type plants (Fig. 11B). These results provide compelling evidence that the overexpression of *DoWRKY71* promotes ABA-mediated stomatal closure in tobacco.

## Expression analysis of *DoWRKY71* gene in transgenic tobacco at cold and ABA treatment

Under cold and ABA treatment, the expression level of this gene initially increased and then decreased, demonstrating a trend of fluctuation. The expression level of treated plants was higher than that of untreated plants (0 h), except for those treated at 4 °C for 1 h; this difference was significant (Fig. 12).

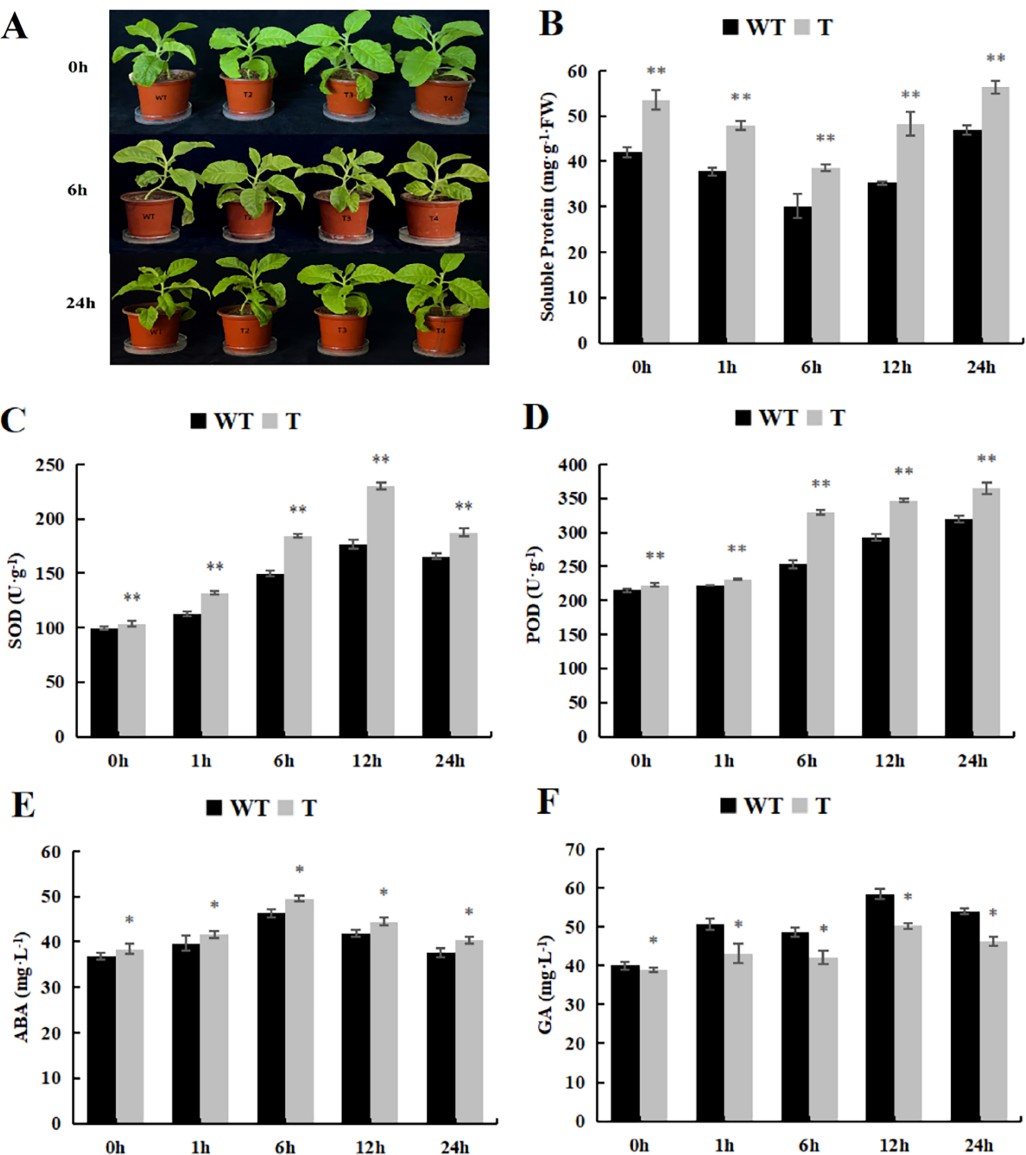

**Figure 10 Changes of physiological and biochemical indexes endogenous and hormone contentunder ABA treatment.** (A) Plant growth; (B) soluble protein; (C) SOD activity; (D) POD activity; (E) ABA content and (F) GA content in the transgenic lines and WT plants under ABA treatment. 0 h: (CK), 6 h: 6 h after treatment, 24 h : 24 h after treatment. WT, wild-type tobacco; T2, transgenic tobacco. Data are presented as mean ± SD (n = 3). * and ** indicate a significant difference at $P < 0.05$ and $P < 0.01$ compared with WT, respectively.

## DISCUSSION

WRKY transcription factors, as a class of regulatory proteins, respond to plant growth by regulating the expression changes of numerous downstream genes (*Wan et al., 2018*). They have been investigated and identified at the genome level and transcriptome level in a variety of plants (*Gu et al., 2018*), which has become a focus of studies on plant gene function. In this study, 25 WRKY gene family members were identified in the transcriptome of *D. opposita*. According to the characteristics of WRKY family and 44 rice

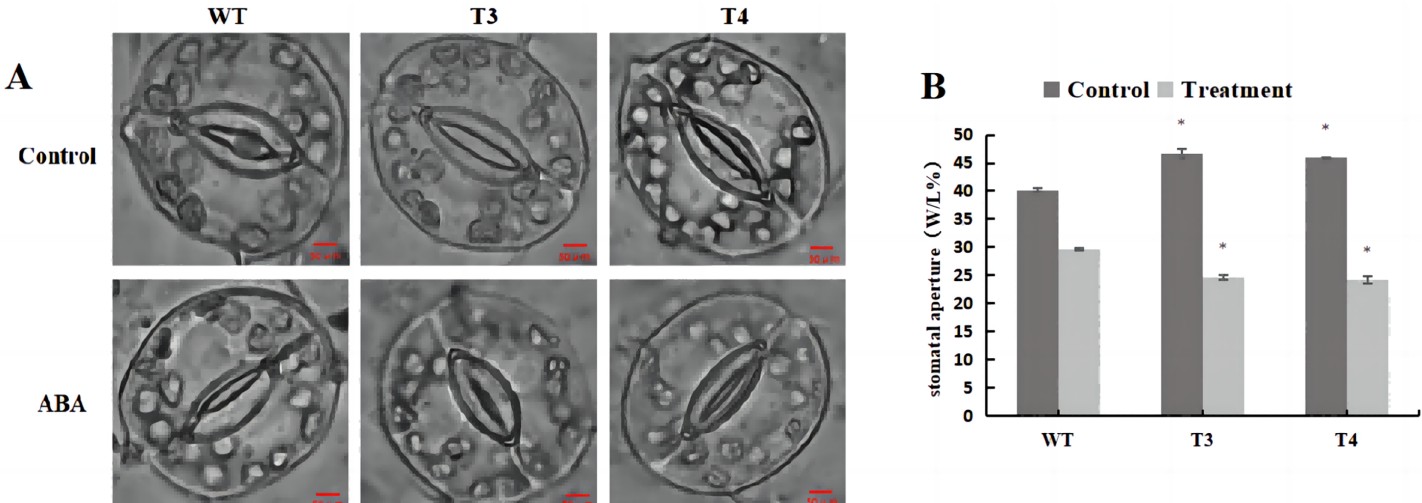

**Figure 11 Stomatal aperture of WT and *DoWRKY71* transgenic lines under ABA treatment.** (A) The stomatal photo of WT and *DoWRKY71* transgenic lines; (B) Stomatal aperture index of WT and *DoWRKY71* transgenic lines. Data are presented as mean ± SD (*n* = 3). * indicate a significant difference at *P* < 0.05 compared with WT, respectively.

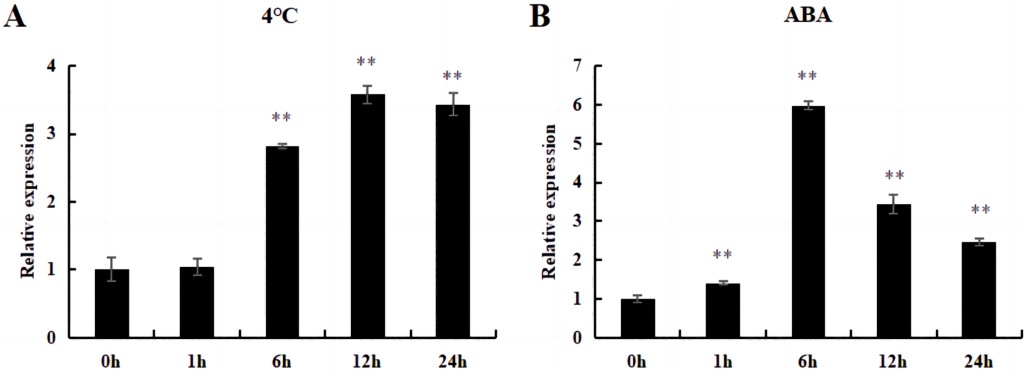

**Figure 12 The relative expression of *DoWRKY71* gene in tobacco leaves under different treatments.** (A) Cold treatment; (B) ABA treatment. Data are presented as mean ± SD (*n* = 3). ** indicate a significant difference at *P* < 0.05 compared with 0 h, respectively.

protein sequences, WRKY proteins of *D.opposita* could be divided into three groups. The WRKY gene family was smaller than that of other plants such as sweetpotatoes (79 WRKY genes) (*Qin et al., 2020*) and eggplants (58 WRKY genes) (*Yang et al., 2020*). Interestingly, the IIb subgroup has no DoWRKY member. This discrepancy might be attributed to the incomplete sequencing of the yam genome and the identification from transcriptome data (*Jiang et al., 2022*). The WRKYGQK heptapeptide domain and zinc finger structure are necessary for binding of WRKY transcription factors to the cis-acting W-box element of target genes. Multiple sequence alignment of DoWRKY proteins shows that all DoWRKY proteins except DoWRKY50 contain WRKYGQK conserved sequences. However, the members of group IIc, such as DoWRKY50, differ by one amino acid, with the glutamine (Q) being substituted by a lysine (K)to form WRKYGKK, a pattern consistent with what is observed in *Lycium ruthenicum* (*Tiika et al., 2020*). It has been

reported that the DNA binding activity of the N-terminal WRKY domain is not as strong as that of C-terminal, and there are more variations in the N-terminal WRKY domain (*Maeo et al., 2001*). The conserved motifs of WRKY exist not only in higher plants such as rice (*Ross, Liu & Shen, 2007*), but also in lower plants such as algae (*Song et al., 2022*), which reveals their extensive involvement in different plants. The identification of conserved motifs in WRKY protein of yam shows that Motif 1 and Motif 2 are the core motifs of yam. These results suggest that these traits are conserved in the WRKY family of yams. It is worth noting that the members of almost every group share similar characteristics. However, studies have shown that members of these families come from the same clade, but their functions are quite different (*Yang et al., 2019*). For example, DoWRKY40a, DoWRKY71a, DoWRKY40b, DoWRKY40c and DoWRKY71b belong to groupIIa, DoWRKY40a, DoWRKY71a, DoWRKY40b, DoWRKY40c contains six conserved motifs, DoWRKY71b contains five conserved motifs, but DoWRKY71a contains only four conserved motifs. Their different conserved motifs may lead to different functions of genes. DoWRKY32 of group I and group IIa all contain conserved motif 5, suggesting that these genes may have similar functions.

Previous studies have shown that members of the WRKY family have diverse functions in plant growth and various stress responses, but the functions in *WRKY* genes within the same group or subgroup usually remain similar (*Chen et al., 2017*, *2022*). Besides, the regulation of root growth is under the control of several hormones and the expression of various transcriptional regulators. Among the expression level of *DoWRKY* genes in *D. opposita*, *DoWRKY21* and *DoWRKY71* were upregulated in the early stages and down regulated in the mature stage. They all belonged to Group II. The results showed that many genes can be divided according to their abundant expressions in specific organs, a finding that might reflect their participation in the common developmental processes (*Sun et al., 2020*). Four genes (*DoWRKY20, -33, -21, -71*) significantly upregulated during cold stress, and their responses are comparable to those of 8 *TkWRKY* genes which were induced by cold stress (*Cheng et al., 2022*). Under ABA stress, four *DoWRKY* genes (*DoWRKY33, -21, -70* and *-71*) were significantly induced. Similar results have been reported in *Scutellaria Baicalensis Georgi* (*Zhang et al., 2022*) and *Gossypium hirsutum* (*Hu et al., 2021*). In addition, the *DoWRKY20, -33, -71, -69* genes are also sensitive to MeJA treatment. Two *DoWRKY* genes (*DoWRKY21* and *-40b*) were down-regulated during MeJA treatment. Research shows that MeJA are identified as a transcription activator of *PpPAL* and *Pp4CL via* binding to their W-box (*Ji et al., 2021*). The above research confirms that the response mechanisms of *DoWRKYs* members may be complex and varied, including cold, ABA, MeJA, and the selection of candidate genes can provide references for future investigations.

Transcriptional regulation of various effects or genes in plants is the key to activating inducible stress responses in plants (*Shekhawat, Ganapathi & Srinivas, 2011*). The sub-cellular location assay displayed that the DoWRKY71 protein was localized in the cell nucleus. WRKY71 localized in the cell nucleus and acted as a promoter of flowering transcriptional regulatory cascadeand in wild strawberries (*Lei et al., 2020*). It is consistent with FvWRKY71, leading to the suspicion that WRKY71 may closely related to the

transcriptional regulatory function in the nucleus. In this study, the transgenic *DoWRKY71* plants were treated with cold and ABA stress, indicating that *DoWRKY71* is sensitive to environmental stress. Chlorophyll, an important part of the plant's photosynthetic apparatus for harvesting light is closely related to the photosynthetic function of the plant and the chlorophyll content of the leaf represents the physiological state of the plant (*Kurniawan et al., 2021*; *Jang et al., 2022*). Under low temperature treatment, the change in chlorophyll content indicated that transgenic plants can improve their photosynthetic capacity by increasing chlorophyll content.

The antioxidant enzyme content is the main index reflecting the degree of peroxidation of cells in the plant (*Zhang et al., 2020*). In this study, under low temperature and ABA treatments, the content of soluble protein first decreased and then increased, the SOD activity first increased and then decreased, and the POD activity showed an overall upward trend. The antioxidant enzyme system protects the membrane system by scavenging reactive oxygen species, thereby alleviating the damage to plants caused by stress (*Marček et al., 2016*). But the effect will weaken over time, and the activity of antioxidant enzymes will gradually decline with the production of reactive oxygen species. Prolonged low temperature can lead to excessive production of reactive oxygen species, resulting in plant cell membrane, protein and nucleic acid damage, and even cell death (*Li et al., 2020*). MDA concentration is a parameter of membrane lipid peroxidation, which can reflect the degree of stress. NBT and DAB staining tests showed that transgenic tobacco suffered lower membrane damage, less accumulation of reactive oxygen species and MDA, and stronger cold tolerance under low temperature stress.

WRKY protein also plays an important role in activating or inhibiting the signalling pathway (*Rushton et al., 2012*), and *White et al. (2000)* found that ABA inhibited GA signal in the early germination of maize. In this study, we observed that endogenous GA and ABA were relatively antagonistic during ABA spraying. Exogenous ABA could reduce leaf stomatal conductance, water loss and the pressure of osmotic regulation system. The leaf stomatal tension decreased under ABA spraying, which facilitates plants' prevention of water loss during adverse conditions. *DoWRKY71* may regulate stomatal opening and closure of plants *via* ABA-dependent signalling pathways, which is consistent with *MfWRKY17* reported in *Argyrophylla* (*Huang et al., 2020*). Tolerance in plants can be manifested in two ways, by enhancing functional gene expression or by promoting protein synthesis. The expression level of transgenic tobacco *DoWRKY71* under both treatments was significantly higher than that of the control, indicating that both low temperature and ABA could trigger the expression of *DoWRKY71* gene in response to various physiological processes. This result is consistent with those of cucumber (*Zhang et al., 2016*) and orchid (*Liu et al., 2021*).

## CONCLUSIONS

In conclusion, this study provided the first comprehensive and systematic analysis of 25 *WRKY* genes identified in *D. opposita* transcriptome. All *DoWRKY* genes can be divided into six groups (categories I–III), and were found to respond to the growth and development process and abiotic stresses. Besides, *DoWRKY71* may be one of the

important members of the WRKY gene family of yam, with the ability to cross-regulate other genes involved in abiotic stress signalling pathways. However, the molecular mechanism of which pathways *DoWRKY71* and other members participate in the growth and development of yam and how it interacts with downstream proteins still needs to be further studied.

### Funding

This work was supported by the National Natural Science Foundation of China Project (No. 32260759); The Inner Mongolia Natural Science Foundation Project (No. 2022MS03052); and the Inner Mongolia Autonomous Region of Basic Research Funds for universities directly project (Special Fund for Seed Industry Revitalization of Leading Talents) (No. BR22-11-06). The funders had no role in study design, data collection and analysis, decision to publish, or preparation of the manuscript.

### Grant Disclosures

The following grant information was disclosed by the authors:
National Natural Science Foundation of China Project: 32260759.
The Inner Mongolia Natural Science Foundation Project: 2022MS03052.
The Inner Mongolia Autonomous Region of Basic Research Funds for universities directly project.
Special Fund for Seed Industry Revitalization of Leading Talents: BR22-11-06.

### Competing Interests

The authors declare that they have no competing interests.

### Author Contributions

- Linan Xing conceived and designed the experiments, performed the experiments, analyzed the data, prepared figures and/or tables, authored or reviewed drafts of the article, and approved the final draft.
- Yanfang Zhang conceived and designed the experiments, authored or reviewed drafts of the article, and approved the final draft.
- Mingran Ge analyzed the data, prepared figures and/or tables, and approved the final draft.
- Lingmin Zhao analyzed the data, prepared figures and/or tables, and approved the final draft.
- Xiuwen Huo conceived and designed the experiments, authored or reviewed drafts of the article, and approved the final draft.

### DNA Deposition

The following information was supplied regarding the deposition of DNA sequences:
The sequences are available at GenBank: OP380252.

## Data Availability
The raw data is available in the Supplemental File.

## Supplemental Information
Supplemental information for this article can be found online at http://dx.doi.org/10.7717/peerj.17016#supplemental-information.

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
