# Peer review of "Identification of WRKY gene family in Dioscorea opposita Thunb. reveals that DoWRKY71 enhanced the tolerance to cold and ABA stress"

_PeerJ, doi:10.7717/peerj.17016_

## Round 0.1 · original submission · Major Revisions

Dear Authors you are advised to revise the manuscript as per reviewers suggestions and submit the revised version.

**Language Note:** The review process has identified that the English language must be improved. PeerJ can provide language editing services - please contact us at copyediting@peerj.com for pricing (be sure to provide your manuscript number and title). Alternatively, you should make your own arrangements to improve the language quality and provide details in your response letter. – PeerJ Staff

Reviewer 1 ·

Basic reporting

It is a well-organized article with a full explanation of figures and data. Although there are some grammer errors, it is easily understood overall.

I suggest the authors request a fluent speaker to proofread the manuscript, to improve the English language. For example,
line 185: it should be “the experiments were …”.
Iine 212: “compared to the groups”.
I didn’t write down other grammar errors. Please double check the language.

Experimental design

It has sufficient details in the methods section. The aim is clearly defined and the data support the conclusion.

Validity of the findings

There are enough replicates for each sample. The findings are supported by the data.

Additional comments

The manuscript entitled “Identification of WRKY gene family in Dioscorea opposite Thunb. Reveals that DoWRKY71 enhanced the tolerance to cold and ABA stress” aimed to study WRKY gene family of Dioscorea opposita Thunb. The authors isolated and identified 25 WRKY proteins in yam from the database, and conducted phylogenetic analysis, gene structure analysis, and conserved domain alignment. Besides, they conducted a compressive analysis of WRKY genes in yam and expression patterns, their responses to abiotic treatments. Among 25 candidate transcription factors, the authors assessed the impact of WRKY71 by overexpression on improved cold and ABA tolerance.

Overall, this study is very interesting. It provides the insights into the potential functions of DoWRKY gene family candidates. Before the acceptance, there are some minor concerns shown as below:


1. Line 151-153: The authors renamed “DoWRKY17” as “DoWRKY71”. I suggest the authors should keep using “DoWRKY71” in the whole manuscript. Otherwise, it is not consistent. Please correct the names in the manuscript.

2. Line 158, 159, 166: “ Agrobacterium tumefaciens ” should be italic.

3. I suggest that the author use “WRKY gene family” in Line 200.

4. I suggest that the author add the refs for the lines 204 – 205.

5. There is no distance scale in the Figure 1.
6. I suggest using two different colors to label DoWRKY and OsWRKY proteins in Figure 1.


7. Line 229 – 230: It seems that 25 DoWRKY gene family proteins evolve to different subgroups. But there is no evidence to show they are independently evolving in each species. Otherwise, please explain and show the evidence.

8. Line 258: it should be “MJA” instead of “JA”

9. Line 378 – 379: it should be “four genes” instead of three genes.

Reviewer 2 ·

Basic reporting

The manuscript titled “Identification of WRKY gene family in Dioscorea opposita Thunb. reveals that DoWRKY71 enhanced the tolerance to cold and ABA stress” by Xing et al., identified and characterize (using a series of in silico analysis) of DoWRKYs family in the D. opposita. The expression pattern of DoWRKY genes at different growth and developmental stages and their response to abiotic treatments were comprehensively analyzed using real-time quantitative PCR (qRT-PCR), providing valuable insights into their unique functions in mediating specific responses. Furthermore, the authors are to be commended for assessing the effect of overexpression of DoWRKY71 on improving cold tolerance and ABA tolerance in tobacco by analyzing the phenotypic and physiological changes in tobacco, as well as the expression of adversity-responsive genes. However, the use of tobacco as a model system may not reflect the actual response of yam to abiotic stresses.
Overall, this study provides meaningful insights into the role of WRKY genes in the response of yam to growth and development and stress-related processes and provides candidates for new avenues of genetic engineering and molecular breeding. But the overall conclusion is not clear. There are still some issues that could be taken care of.

Experimental design

1. “By comprehensively considering the protein family classification and subcellular localization, we selected 8 genes from six groups to analyze the expression level of yam under different growth and development stages.” What basis 8 DoWRKYs out of a total 25 DoWRKYs was selected? Please provide details, in Supplementary information.

Validity of the findings

1. Apart from the general characterization, what specific information was drawn from the amino acid length, molecular weight (MW), and isoelectric point (pI) of all candidate DoWRKYs. Please identify the top-performing genes among the whole candidate DoWRKYs.
2. In lines 251-252, it is stated that "DoWRKY12 was up regulated in the whole process and highly expressed in 165 d." However, in Fig. 5, it is shown that DoWRKY12 was down-regulated at 105d compared to 90d. 105d compared to 90d is down-regulated.
3. In lines 263-265, it is stated in the text: “The expression levels of DoWRKY8 and DoWRKY9 genes showed an overall upward trend, reaching the highest level at 12h and becoming stable”. However, in DoWRKY8 shown in Fig.6(A), the expression was significantly reduced by 24 h of low temperature stress, not stabilized.
4. The description of the conclusion is not clear in lines 268-269 “The expression levels of DoWRKY7 and DoWRKY16 were eight-fold higher than those of the control, respectively.”
5. Please reconfirm whether the significance labeling of the bar graphs in Fig.9 and Fig.10 is correct. For example, Fig.9 (C) whether there is a significant difference between treatment and control at 12h of stress.
6. Please justify the sentence Discussion#365-367 “For example, DoWRKY10, DoWRKY11, DoWRKY12, DoWRKY13 and DoWRKY15 belong to group II, but have similar conserved motifs to which group I.”
7. Please provide references for the discussion #356-358 “It has been reported that the DNA binding activity of the N-terminal WRKY domain is not as strong as that of C-terminal, and there are more variations in the N-terminal WRKY domain.”

Additional comments

1. Please provide clear images of Fig.1 - Fig.11 in the figure files section.
2. Fig.5 missing units in the horizontal coordinates.
3. Fig. 12 was not found in the submitted manuscript.
4. Please standardize the style of all references, as some authors' names are abbreviated, and some are full names.

Reviewer 3 ·

Basic reporting

Authors have used an integrated bioinformatics and experimental pipeline for the identification of WRKY gene family in Dioscorea opposite Thunb, and defined the role of DoWRKY71 in enhancing the tolerance to cold and ABA stress. The study has been performed well and the manuscript is also well written.

Experimental design

1. It would be better to provide the whole analysis as a flowchart in image.
2. In bioinformatics section, not much details of tools/software are provided.
3. It would be intersecting to see which genes DoWRKY71 is regulating. The data could be downloaded from TF-gene/protein databases.
4. A more understanding may be obtained if you check for the pathways/processes these genes (point 3. above) are involved in.

Validity of the findings

no comment

Additional comments

Check the manuscript for grammatical errors and typos.
Improve the image quality.

---

## Round 0.2 · Minor Revisions

Please address the issues raised by reviewers.

Reviewer 1 ·

Basic reporting

It is clear and unambiguous. The authors have improved the language in the revised manuscript.

Experimental design

Methods are described thoroughly in the revised manuscript.

Validity of the findings

The authors made a substantial improvement in the revised manuscript.
Conclusions are well stated.

Additional comments

No comments

Reviewer 2 ·

Basic reporting

no comment

Experimental design

no comment

Validity of the findings

no comment

Additional comments

I have gone through the revised version of your manuscript. Thank you for all of the excellent work by the authorship team that has obviously gone into the generation of a much improved version of your originally submitted, and peer reviewed manuscript.
However, a number of references issues remain which now need to be addressed. There are issues with inconsistent formatting of references, such as years and authors.

Reviewer 3 ·

Basic reporting

The authors have not answered many of my major questions, as they plan to do analyses in later experiments.

Experimental design

NA

Validity of the findings

NA

Additional comments

NA

---

## Round 0.3 · Minor Revisions

Dear Authors,

Unfortunately, your manuscript still requires further improvement before acceptance. Please, proofread it accordingly, including spacing (as it cannot be accepted until proper formatting and language have been assured).

Methods are not complete enough to be reproducible. They should include the kits' actual commercial names and references/cat numbers. For example, for the cDNA kit, I found that the company you cite has multiple identical products. Using references to indicate the methods is not a good practice. You should also briefly describe what you have done. Particularly in key steps. This sometimes is cared for, but not always, which also can induct one to think that several different writers put the manuscript together not assuring for consistency.

Figure 12 (eg) legend mentions *=p<0.05 but none is in the graphs. while in figure 11 is with **<0.01. I don't find it in the figures. This shows a lack of professionalism. I think figure quality or readability is not good enough either sometimes., so please, ensure the submission of quality figures as per guidelines.

---

## Round 0.4 · accepted · Accept

Dear authors, I am happy to let you know that I am accepting your manuscript for publication in PeerJ. congratulations! The production will be in touch with you. I noticed that some things still require careful proofreading, so I'm taking this opportunity to highlight the importance of proofreading and triple-check your data presentation/ statistical results, in particular, so that you make sure that upon publication all is transparent, accurate, and precise and readily reflects the quality of this important work. All the best.